# Development of Polymeric Membranes for Oil/Water Separation

**DOI:** 10.3390/membranes11010042

**Published:** 2021-01-08

**Authors:** Arshad Hussain, Mohammed Al-Yaari

**Affiliations:** Chemical Engineering Department, King Faisal University, P.O. Box 380, Al-Ahsa 31982, Saudi Arabia; ahussin@kfu.edu.sa

**Keywords:** oil–water separation, polymeric membrane, cellulose acetate, Nylon 66, permeability

## Abstract

In this work, the treatment of oily wastewater was investigated using developed cellulose acetate (CA) membranes blended with Nylon 66. Membrane characterization and permeation results in terms of oil rejection and flux were compared with a commercial CA membrane. The solution casting method was used to fabricate membranes composed of CA and Nylon 66. Scanning Electron Microscopy (SEM) analysis was done to examine the surface morphology of the membrane as well as the influence of solvent on the overall structure of the developed membranes. Mechanical and thermal properties of developed blended membranes and a commercial membrane were examined by thermogravimetric analysis (TGA) and universal (tensile) testing machine (UTM). Membrane characterizations revealed that the thermal and mechanical properties of the fabricated blended membranes better than those of the commercial membrane. Membrane fluxes and rejection of oil as a function of Nylon 66 compositions and transmembrane pressure were measured. Experimental results revealed that the synthetic membrane (composed of 2% Nylon 66 and Dimethyl Sulfoxide (DMSO) as a solvent) gave a permeate flux of 33 L/m^2^h and an oil rejection of around 90%, whereas the commercial membrane showed a permeate flux of 22 L/m^2^h and an oil rejection of 70%.

## 1. Introduction

Industrialization has given birth to many problems like increased greenhouse gas emissions, environmental pollution, and energy crises [1,2,3]. Especially, water is indispensable for the existence and survival of life on earth. In the past few years, the world has faced severe scarcity in water resources. Extensive use of water, extreme drought periods, upsurge in population, and water pollution [4] have resulted in a global concern about secure drinking water supply. Polluted water is a threat to the life of humans, animals, and plants both ashore and in the oceans. The situation will get worse if timely actions are not taken to combat the issue of water pollution, which would be a hindrance for the sustainable development of society [5]. Therefore, clean drinking water and sanitation are among the seventeen Sustainable Development Goals of the United Nations Environment Programme (UNEP) [6]. According to the United Nations World Water Development (UNWWD) report, about 748 million people do not have access to clean drinking water. In addition, by 2050, the demand of water in industry will increase to 400 percent [7], thus aggravating the water crisis even further. In developing countries, 3.2 million children die annually due to poor sanitation and unsafe drinking water [8]. This daunting situation has urged society and especially scientists to explore efficient and sustainable technologies and materials (i.e., such which can be justified from an ecological and economical viewpoint to be applied by and for society now and in the future) for the mitigation of water pollution.

In a variety of industrial processes, water comes into contact with several pollutants such as hydrocarbons, hazardous chemicals, sewage sludge from boilers, cooling towers, and heat exchangers [9]. Industrial wastewater is usually not reusable and may have drastic impacts on living species [10,11]. The wastewater coming from oil processing facilities poses a serious threat to aquatic life and pollutes groundwater [9]. Recycling of polluted water or reduction of pollutants concentration to an acceptable level is considered to be the only way out in these grim circumstances. Conventional techniques including decanting, biological treatment, chemical treatment, gravity separation, skimming, air flotation, coagulation, and flocculation are being used for the treatment of oily wastewater [12,13]. Such processes require some chemicals and biological solvents to treat wastewater which could cause devastating effects on mankind [12]. High cost, use of toxic compounds, and large space requirements are some other drawbacks that limit their usage as suitable techniques [12]. Hence, there is a need to develop suitable treatment processes that produce less hazardous pollutants.

Membrane technology offers great potential to treat oily wastewater. In this context, a range of polymer membranes have been prepared using various techniques [14,15] and the feasibility of a membrane-based oil–water separation process has been reported by many researchers [16,17,18,19]. The cellulose-acetate-based membranes often show high hydrophilicity, high water permeability, and low membrane fouling tendency [20,21].

Li et al. [13] developed a hydrophilic cellulose acetate (CA) membrane for oil–water separation. The membrane was comprised of CA/monohydrate/N-methyl morpholine-N-oxide (NMMO.H_2_O)/polyethylene glycol (PEG 400). Membrane was tested for pure water flux first and then tested for the separation of oil and water. Fouling resistance of the membrane was also measured and analyzed using osmotic-pressure-adsorption model. Over 99% of oil retention was reported and total oil content in the permeate was found to be 10 mg/L. Flux reduction was noted and dominated by concentration polarization.

In order to get a high permeation flux with high antifouling property, a new membrane was developed which was composed of CA grafted with polyacrylonitrile [22]. The new membrane was tested for flux and antifouling property, and results were very promising.

Yan et al. [23] modified polyvinylidene fluoride-based membranes with nanosized inorganic alumina particles in order to purify the oily wastewater stream. Results obtained after the processing of the membrane revealed that the oil content in the permeate stream was reduced to 1 mg/L. Suspended particles in the permeate stream were also reduced to less than 1 mg/L, which is desirable by oilfield drainage.

In 2012, Kota et al. [24] reported the membranes having hygro-responsive properties for oil–water separations. In their study, poly (ethylene glycol) diacrylate (PEGDA) and fluoro-decyl polyhedral oligomeric silsesquioxane (FPOSS) were used as substrate materials. The results demonstrated that the membranes are both super hydrophilic and super oleophobic. A separation efficiency of greater than 99% was achieved. After 100 h of operation, the water flux was nearly 210 L/m^2^ h.

Mansourizadeh et al. [25] developed a membrane composed of polyethersulfone (PES) and CA. Polyethylene glycol (PEG400) was also added into the membrane dope to increase the number of pores in the membrane. CA was added into PES to increase the overall hydrophilicity of the membrane and to gain the desired structure for oil–water separation. Hydrophilic PES/CA membrane showed an increase in water flux up to 27 L/m^2^ s and oil rejection of 88%. This study is aimed at further improving the performance of cellulosic membranes for oil–water separation.

Recently, antibacterial nanofiber membranes for potential application for oil–water emulsions has been reported by Mousa et al. [26]. Polysulfone (PSF) and CA were used as base polymers. Zinc oxide (ZnO) nanoparticles were also incorporated to enhance the properties of the polymeric membranes. The membranes were modified with NaOH and showed better properties than unmodified ones. The highest reported water flux was 420 L/m^2^h.

The objective of this research work is to synthesize and characterize polymeric membrane comprising of CA and nylon 66 blends in different solvents. The effect of solvent on the overall morphology and structure of the membrane is studied. Mechanical, thermal, and morphological properties of the membrane were characterized using tensile testing machine, thermogravimetric analysis (TGA), and Scanning Electron Microscopy (SEM), respectively. Overall permeate flux and rejection of oil were determined as functions of nylon 66 concentration. In addition, the effect of the transmembrane pressure on the overall flux and oil rejection was examined using the permeation experiment setup.

## 2. Materials and Methods

### 2.1. Materials

CA with a molecular weight of 30,000 g/mol was acquired from Sigma-Aldrich (St. Louis, MO, USA). CA was available in a powder form and no pretreatment was required for its further usage. Formic acid was also procured from Sigma-Aldrich (St. Louis, MO, USA) having a purity of 98%. Nylon 66 was purchased from Zytel Dupont (Pulau Sakra Island, Singapore) and was available in a pellet form. Dimethyl Sulfoxide (DMSO) was procured from Sigma-Aldrich with a purity of 99% and a transparent color. Commercial CA membrane with a pore diameter of 0.22 µm was purchased from Merck Millipore (Boston, MA, USA).

### 2.2. Preparation of Polymeric Membranes

Polymer solutions of different wt% of CA and Nylon 66 in DMSO and Formic Acid were cast on a glass plate at room temperature. Membrane thickness of 0.3 mm was maintained with the help of a doctor blade. Solvents were first evaporated from the casted membrane and then immersed into nonsolvent (water). After immersion, the film color changed from clear to white and also detached from the glass plate. Membranes were collected from the coagulation bath and preserved in plastic bags for the characterization and permeation tests. Table 1 and Table 2 show the composition of membrane solution using DMSO and Formic Acid, respectively.

### 2.3. Scanning Electron Microscopy (SEM)

To investigate the morphology of the membrane, dry polymeric membrane was dipped in liquid nitrogen and then fractured in order to get the fibrous cross-section of the membrane. Before the analysis, the surface of the samples was sputter coated with a thin layer of gold to avoid the accumulation of charges. An acceleration voltage of 20 kV was used for investigation. For the cross-section analysis, the formulations were soaked in liquid nitrogen to freeze crack. Afterwards, they were also sputter coated with gold. Surface morphology and cross-section of the membrane were then studied using JEOL JSM-6490A SEM (Tokyo, Japan).

### 2.4. Permeate Flux Determination

A crossflow permeation testing unit, shown in Figure 1, was used to evaluate the overall flux across the membrane. The permeation testing unit consists of a feed tank, permeate collector, and centrifugal pump. The effect of transmembrane pressure was recorded with the help of a ball valve mounted on the retentate stream.

An oil–water emulsion was prepared by mixing lubricating oil with water in different proportions at 900 rpm for 30 min. The membrane was washed with distilled water before determining the permeate flux. The membrane was placed on a porous Teflon support to prevent bending at various feed pressures. The total membrane area exposed to the feed was about 0.0094 m^2^. Flux was calculated by varying the transmembrane pressures (0.5, 1.0, and 1.5 bar). Permeate flux was calculated using the following equation.
(1)$$J_{w}=\;\frac{Q}{A\;\times t}$$
where

*J_w_* denotes the permeate flux (mL/m^2^ h),

*Q* denotes the permeate volume (mL),

*A* denotes the membrane area exposed to the feed (m^2^), and

*t* is the sampling time for permeate (h).

### 2.5. Oil Rejection

Percentage of oil rejection (R%) was calculated using the same method as used for oil and grease calculation. Concentrations of oil in feed and permeate were calculated using n-Hexane and HCL. The rejection (%) was calculated using the following equation:(2)$$R\;\%=\left(1-\;\frac{C_{p}}{C_{f}}\right)\times 100$$
where

*c_p_* denotes the permeate concentration (g/L),

*c_f_* denotes the feed concentration (g/L).

### 2.6. Thermal Studies

Thermogravimetric analysis (TGA) was carried out by using Perkin Elmer S11 diamond TG/DTA (Boston, MA, USA). The sample weighing 3 mg was dried at a heating rate (β) of 15 °C/min from 300 °C to 600 °C in an inert atmosphere.

### 2.7. Mechanical Properties

Mechanical properties of the membrane were tested using a Universal Testing Machine (ELE International, Loveland, CO, USA). The mechanical analysis of polymeric membranes was carried out according to the ASTM D638 standard. All the specimen were cut into lengths of 50 mm and widths of 15 mm. The crosshead speed was 50 mm/min. The thickness of each sample was 0.3 ± 0.05 mm. The tensile strength (MPa) and elongation at break (%) were measured at room temperature.

## 3. Results and Discussion

### 3.1. Morphological Analysis of Membranes

Morphology of a commercial membrane and the developed membranes were analyzed by SEM. Cross-section, outer surface, and skin layer morphologies of the polymeric formic acid-based and DMSO-based membranes are shown in Figure 2 and Figure 3, respectively. As shown in Figure 2, the commercial membrane (C1) has a homogeneous spongy structure with no skin layer on the surface.

Solubility between solvents and nonsolvents is a critical factor in membrane synthesis. Solvents dissolve a wide variety of polymers and, based on their solubility parameter, they give porous and anisotropic membranes. The solubility parameter is a numerical value that indicates the relative solvency behavior of a specific solvent [15]. Generally, polar solvents like DMSO and Formic Acid are considered to be the best solvents for casting CA membranes as they precipitate rapidly when immersed in water and thus produce anisotropic membranes with high pore density and high flux. As reported, DMSO has the lowest water contact angle but the highest hydrophilicity [27]. Therefore, in this work, DMSO and Formic Acid have been used as solvents.

In most of the cases, a membrane having fingerlike voids along its cross-section (F1, F3, D1, and D3) is preferable than having a spongy cross-section (C1) in the case of oil–water treatment [22] (see Figure 2 and Figure 3). Therefore, the developed membranes are preferable over the commercial membrane.

In addition, SEM results showed that Nylon 66 was homogeneously dispersed on the membrane surface (Figure 2). The presence of Nylon 66 on the membrane surface increased its surface roughness whereas the cross-sectional analysis of the membrane confirmed its asymmetrical structure. The compact structure and the decreasing porosity of the support layer across its cross-section can be attributed to the affinity of solvent with nonsolvent [28]. The affinity of Formic Acid was low, which resulted in the compact structure of the membrane (Figure 2).

Nylon is a natural hydrophilic polymer and has a wide range of compatibility and resistance to organic solvents. The effect of nylon concentration on the overall morphology of the membrane was also examined by SEM. Results showed that as the concentration of nylon increased from 2 wt% to 5 wt% in the polymer solution, the roughness of the surface increased as shown in Figure 2 and Figure 3. An increase in the Nylon 66 concentrations also affects the cross-sectional morphology of the membrane. The thickness of the skin layer increases as the amount of Nylon 66 is increased. The thickness of the skin layer is inversely proportional to the permeate flux and hence, permeate flux decreases as the amount of Nylon 66 increases in the casting solution.

Moreover, the cavities available along the cross-section of the membrane also ceased to exist. The same effect was noted by Young et al. [29]. An increase in the polymer amount resulted in a slow liquid–liquid demixing which causes a decrease in porosity and pore-to-pore distance was increased leading to less porosity of the membrane [30].

SEM surface images revealed that nylon is finely dispersed in CA, which confirms their compatibility with each other. Moreover, the hydrophilicity of the membrane surface has been enhanced by the addition of Nylon 66, as indicated by the increase in the oil rejection rate.

Pores were also available at the surface, which confirmed the ultrafiltration structure [27]. Cavities were available across the cross-section of the membrane separated by a honeycomb structure (Figure 3). In addition, Figure 3 shows the asymmetrical structure of the membrane which is also confirmed by other researchers [29]. The porosity of the skin layer, as well as the availability of cavities across the cross-section of the membrane, could be justified by the fact that DMSO has a higher affinity with nonsolvents (water in this case). Due to the high solubility parameter of DMSO, the polymer membrane is relatively porous (see Figure 3) compared to the membrane formed by using Formic Acid (see Figure 2). Less nonsolvent dissolved in a high amount of solvent resulted in the production of cavities across the cross-section and pores on the surface.

This phenomenon has also confirmed that the affinity of solvent with nonsolvent results in the production of fingerlike cavities as well as thin skin layer [29]. Similar behavior of Nylon 66 was observed by Lin et al. [31]. Nylon has become more prominent on the surface of membranes. The surface of the membrane had porosity owing to the lesser width of the skin layer. The addition of Nylon 66 results in an increase in the solution’s viscosity [28]. In the case of a lower concentration of Nylon 66, large cavities were seen as shown in Figure 2. As the amount of nylon increases in polymer solution, the number of cavities was replaced with solid material having less porosity [30].

### 3.2. Thermogravimetric Analysis (TGA)

TGA was performed to analyze the thermal stability of the developed membrane (Figure 4). The TGA curve showed three zones of weight loss at the temperature ranges of below 100 °C, 320–400 °C, and above 400 °C. In the case of the developed membranes, the first drop in the TGA curve below 100 °C is due to the loss of moisture. Major weight loss of the membrane was noticed between 320–400 °C. A drop in the curve would be the result of the degradation of CA [32]. The last drop in the TGA curve starting at 400 °C indicated the overall decomposition of the membrane.

TGA of CA-based commercial membrane (C1) was also performed to compare its thermal properties with those of developed membranes at the same conditions. TGA curve of the C1 membrane showed that a major drop in the curve was noticed at about 180 °C and it continues until 250 °C, where complete degradation of the membrane was observed.

TGA results revealed that the developed membranes (D1 and F1) are more resistant to high temperatures and showed less mass loss even at high temperatures when compared to the commercial membrane (C1). This may be attributed to the formation of extra hydrogen bonding interactions between CA and Nylon 66. Consequently, extra energy was required to break these additional bonds [33]. Moreover, the solvent has no significant effect on the thermal properties of the membrane. This finding also suggests that both polymers are compatible with each other and the addition of Nylon 66 enhances the thermal properties of CA [34].

### 3.3. Tensile Testing of Membranes

Polymer membranes must have the necessary mechanical properties to withstand the pressure gradient which acts as a driving force for the separation process [35]. Symmetrical or dense membranes have better mechanical properties when compared with asymmetric membranes. Asymmetrical membranes generally have a skin layer and large cavities which deteriorate the mechanical strength of the membrane. However, symmetrical membranes are compact, do not have fingerlike voids, and provide good mechanical properties [35]. Both developed and commercial membranes were tested to compare their mechanical properties and the average results are presented in Table 3 and Figure 5. It was observed that commercial membrane (C1) shows high tensile strength up to 9 MPa. SEM image of the commercial membrane (Figure 2) revealed its compact, symmetrical structure along the cross-section which results in high tensile strength. On the other hand, elongation at the break of the commercial membrane was about 2.9% which may be because of its porous structure. The porous structure of the membrane reduces both the overall flexibility and elongation at break.

Developed membrane D1 (CA with 2 wt% Nylon 66 blended membrane having DMSO as a solvent) was also tested for its satisfactory mechanical properties. D1 membrane showed a tensile strength of 6.9 Mpa and elongation at break of about 2.5%. It is also reported in the literature that the tensile strength of CA membrane, having an asymmetrical structure, has a maximum tensile strength of 6.9 MPa [35]. SEM images of the D1 membrane (Figure 3) show that the membrane had an asymmetrical structure with large cavities which results in a decrease in elongation.

Developed membrane F1 (CA with 2 wt% Nylon 66 blended membrane having Formic Acid as the solvent) was also tested for its mechanical properties and results were very promising. These membranes showed a tensile strength of about 14.3 MPa and elongation at break of approximately 6%. SEM images revealed that Nylon 66 was homogeneously dispersed on the surface of the membrane. Such a homogeneous dispersion of nylon in the CA matrix imparts the inherent mechanical properties of nylon to the overall properties of the membrane. Membrane has fingerlike cavities beneath the skin layer, but these cavities are not too wide. Moreover, the cavities extend to a porous, spongelike structure which provides more elongation without any breakage in the membrane. It was observed that as the amount of Nylon 66 increases in polymer solution, tensile strength as well as elongation at break increases. This increase in mechanical properties may be due to more interaction between polymeric chains. Moreover, the membrane structure also changed from porous to dense which resulted in an increase in the values of mechanical properties [35], as shown in Figure 5.

### 3.4. Flux

Permeate flux of commercial and developed membranes was determined by varying transmembrane pressure and Nylon 66 composition.

#### 3.4.1. Effect of Composition on the Permeate Flux

The composition of Nylon 66 in polymer solution was varied to evaluate the effect of nylon on membrane flux. Results showed that as the composition of Nylon 66 increased, the membrane permeate flux decreased. As the amount of Nylon 66 increased in the polymeric solution, cavities were filled up along the cross-section of the membrane. Moreover, a higher amount of Nylon 66 increases the surface roughness which results in more adhesion of oil droplets over the membrane surface and the formation of a resistant layer which may hinder the contact of CA with water at the surface and thus results in an overall reduction of flux. Both membrane materials (CA and Nylon 66) swell when exposed to water [36,37]. The polymer swelling reduces the overall porosity of the membrane. As a result, a compact structure, having less porosity and ultimately less flux, was formed. Figure 6a and Figure 7a show the behavior of flux as a function of time for various concentrations of Nylon 66 in the case of D- and F-type membranes, respectively.

#### 3.4.2. Effect of Transmembrane Pressure on Flux

The effect of transmembrane pressure (TMP) on the overall flux of the membrane was also studied. Results for both D- and F-type membranes showed that an increase in transmembrane pressure resulted in an increase in the overall flux across the membrane which may be attributed to the low thickness of the skin layer. An increase in the transmembrane pressure may cause some deformation in polymeric chains as well as in pores’ dimensions [38]. More permeate would pass through the voids as a result of chain movement. However, over time, membrane flux decreases because oil droplets that managed to pass through the surface of the membrane later block the pores available on the surface and throughout the cross-section. Moreover, the overall flux of the membrane decreases as a result of the formation of a resistant layer on the surface of the membrane.

However, owing to the greater thickness of the skin layer of F-type membranes (than that of the D-type membrane), the change in flux was not significant. Figure 6b and Figure 7b show the effect of transmembrane pressure on the overall flux of the D- and F-type membranes, respectively.

### 3.5. Rejection of Oil

Permeate was treated in order to calculate the amount of oil that manages to pass through the membrane. The oil and grease hexane extraction method was used to examine the amount of oil.

#### 3.5.1. Oil Rejection as a Function of Composition

Oil rejection was calculated by varying the composition of the polymer solution. The amount of Nylon 66 in polymeric solution was varied and its effect on oil rejection was studied. Results showed that as the amount of nylon increases, oil rejection increases as well. This increase in oil rejection could be attributed to a change in the membrane structure due to the addition of Nylon 66. While the overall membrane porosity decreased, the thickness of the skin layer increased and, thus, more oil is rejected. Moreover, Nylon 66 offers a barrier to oil and any increase in the amount of nylon in polymeric solution decreases the amount of oil in the permeate. Figure 8a and Figure 9a show the oil rejection of the D-type and the F-type membranes, respectively, as a function of time for various Nylon 66 concentrations at constant pressure of 0.5 bar.

#### 3.5.2. Oil Rejection as a Function of Transmembrane Pressure

Oil rejection was also calculated as a function of transmembrane pressure to examine the effect of pressure on membrane performance. Results showed that as transmembrane pressure across the membrane increases, the overall oil rejection decreases. This behavior of the membrane could be due to pore blockage or the formation of a resistant layer over the surface of the membrane [36]. An increase in transmembrane pressure across the membrane resulted in a higher passage of oil droplets across the membrane. However, stiffness imparted by Nylon 66 to the membrane reduced the effect of high pressure on the membrane [34] and oil rejection of membrane did not rapidly decrease. However, it is worth mentioning that oil rejection was higher in the case of F-type membranes than D-type membranes. This increase in oil rejection might be due to the high miscibility of Nylon 66 in Formic Acid. Figure 8b and Figure 9b show the effect of transmembrane pressure on the oil rejection through all tested membranes.

Experimental results in terms of permeate flux and oil rejection have been compared with the results of other research groups (Table 4). Developed membranes have the potential to be used for oil/water separation. It is evident from the comparison that membrane has a higher oil rejection with modest water flux. It is also worth mentioning here that the polymer used is CA, which is among the cheapest polymers and has the potential to make the process cost-effective for large scale applications.

## 4. Conclusions

In this work, cellulose acetate membranes reinforced with Nylon 66 were successfully fabricated with the solution casting technique. The developed membranes were subjected to extensive characterization in terms of surface analysis, flux permeation, oil rejection, thermal analysis, and mechanical analysis. In comparison to commercial pristine CA membrane, a significant improvement in properties was found. Scanning electron micrographs reveled the uniform dispersion of Nylon 66 into the polymer matrix, this can be a reason for the enhancement of mechanical and permeation properties. The onset of backbone degradation started at 180 °C in case of pure CA membrane. However, it was shifted to 320 °C after the addition of 2% Nylon 66 in CA matrix. The results confirmed the improvement in thermal properties after the addition of Nylon 66 into CA matrix. The tensile strength of the composite membranes improved drastically after the addition of Nylon 66 when formic acid was used as solvent. A similar trend was found for elongation at break. The maximum tensile strength and elongation at break of 36 MPa and 7%, respectively, were depicted by F3 membrane. The results encourage the application of prepared membranes at relatively high pressures. Likewise, the flux permeation and oil rejection were significantly higher than pure CA membrane. As the transmembrane pressure increased, the flux increased accordingly. Permeate flux of 33 L/m^2^h and oil rejection of 95% were achieved by the membrane developed in this work. All these results confirm the effectiveness of CA membranes reinforced by Nylon 66 for oil–water separations.

## Figures and Tables

**Figure 1 membranes-11-00042-f001:**
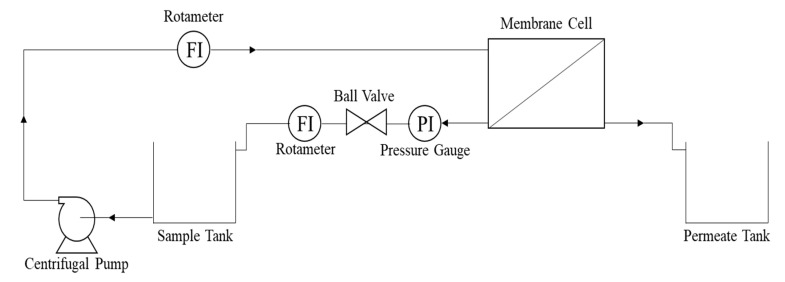
Schematic diagram of the membrane testing unit.

**Figure 2 membranes-11-00042-f002:**
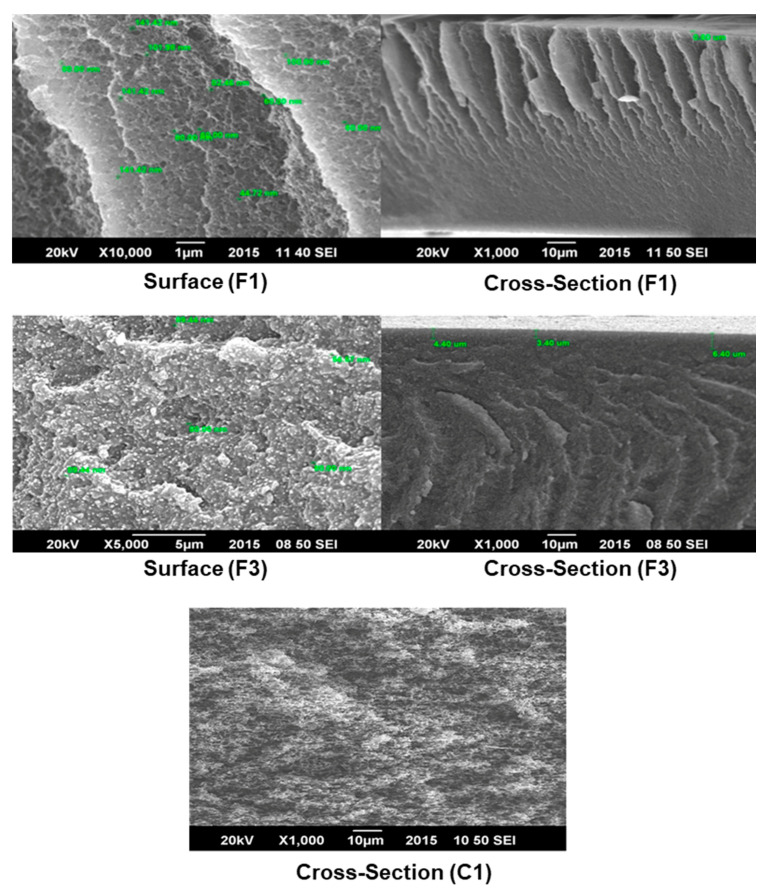
SEM images of surface and cross-section of the Formic-Acid-based membranes (F1 and F3) and commercial membrane (C1).

**Figure 3 membranes-11-00042-f003:**
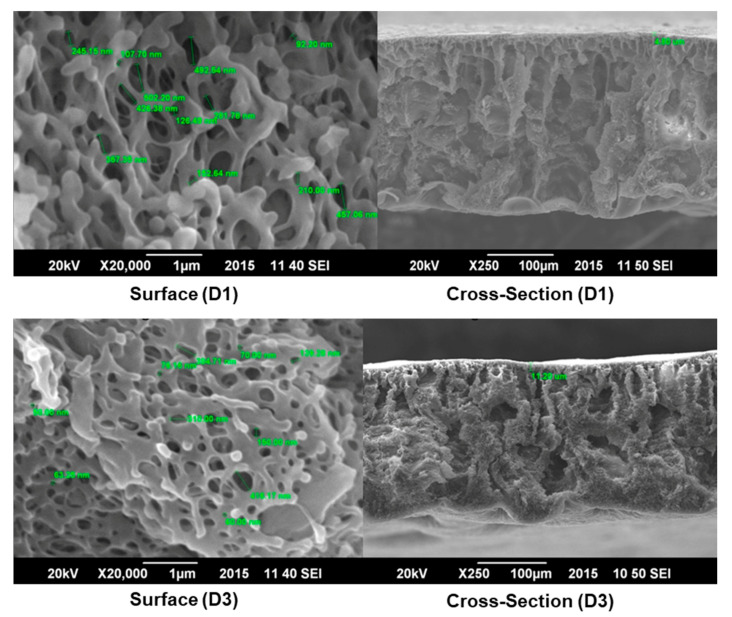
SEM images of surface and cross-section of DMSO-based membranes (D1 and D3).

**Figure 4 membranes-11-00042-f004:**
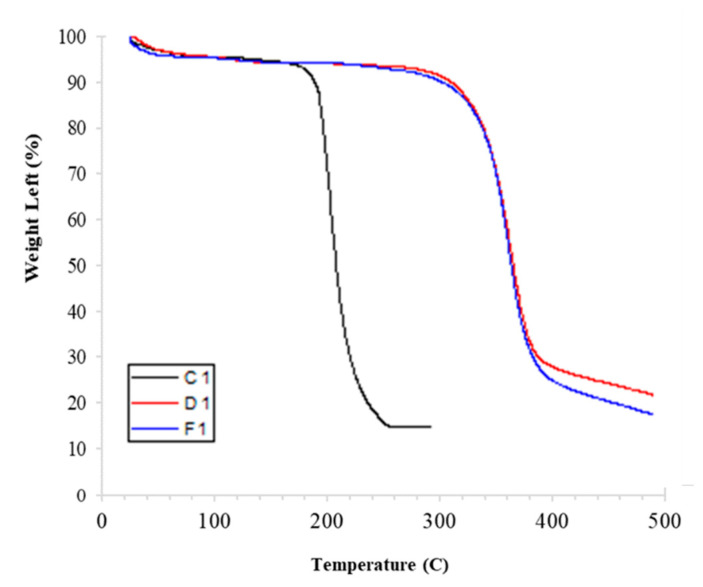
Thermogravimetric analysis (TGA) curves of different used membranes.

**Figure 5 membranes-11-00042-f005:**
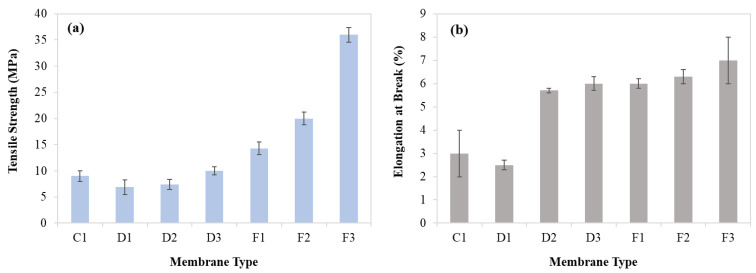
Mechanical properties of the fabricated membranes. (**a**) Tensile strength; (**b**) elongation at break (%).

**Figure 6 membranes-11-00042-f006:**
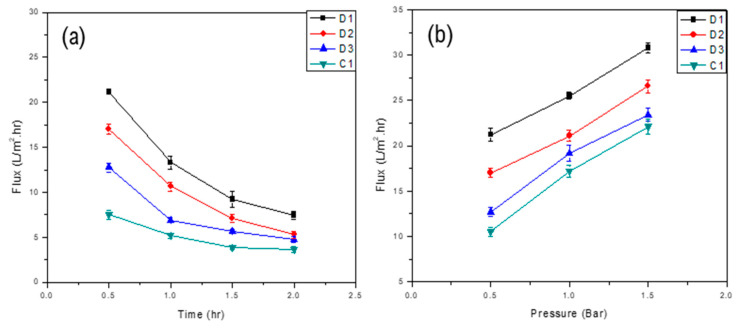
Flux of D-type membranes as a function of (**a**) time and (**b**) transmembrane pressure.

**Figure 7 membranes-11-00042-f007:**
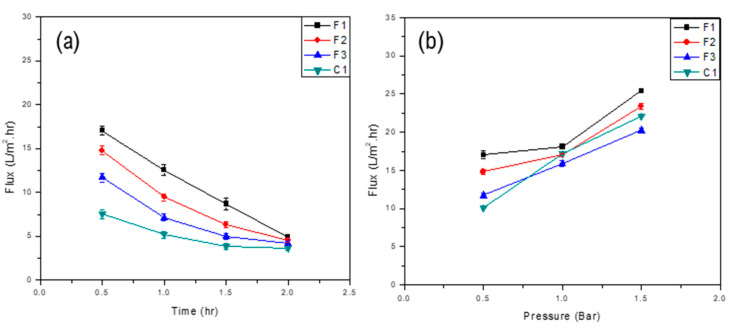
Flux of F-type membranes as a function of (**a**) time and (**b**) transmembrane pressure.

**Figure 8 membranes-11-00042-f008:**
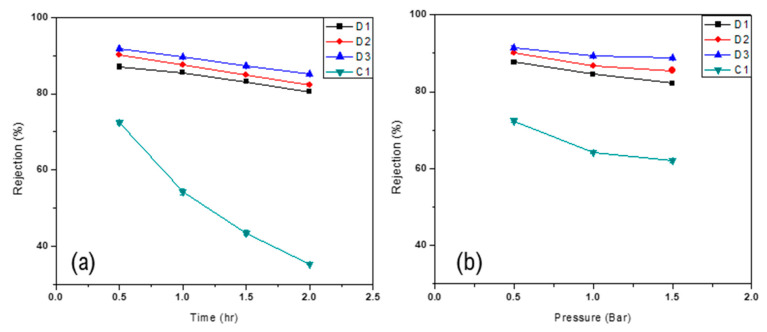
Oil rejection of D-type membranes as a function of (**a**) time at 0.5 bar and (**b**) pressure.

**Figure 9 membranes-11-00042-f009:**
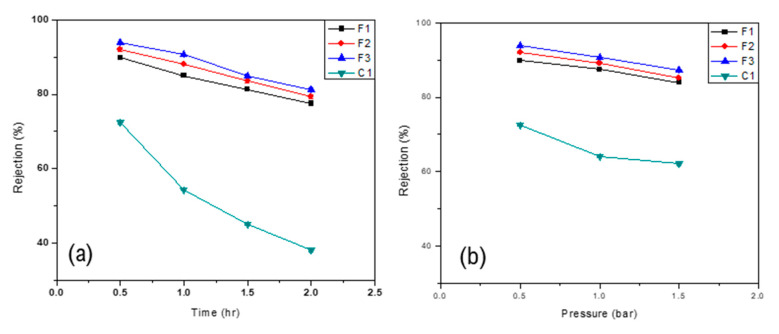
Oil ejection of F-type membranes as a function of (**a**) time at 0.5 bar and (**b**) pressure.

**Table 1 membranes-11-00042-t001:** Composition (wt%) of the developed membrane using Dimethyl Sulfoxide (DMSO) as a solvent.

Solution	Cellulose Acetate (wt%)	Nylon 66 (wt%)	DMSO (wt%)
D1	11	2	87
D2	11	3	86
D3	11	5	84

**Table 2 membranes-11-00042-t002:** Composition (wt%) of the developed membrane using Formic Acid as a solvent.

Solution	Cellulose Acetate (wt%)	Nylon 66 (wt%)	Formic Acid (wt%)
F1	11	2	87
F2	11	3	86
F3	11	5	84

**Table 3 membranes-11-00042-t003:** Tensile testing of membranes.

Membrane	Tensile Strength (Mpa)	Elongation at Break (%)
C1	9	3
D1	6.9	2.5
D2	7.4	5.7
D3	10	6
F1	14.3	6
F2	20	6.3
F3	36	7

**Table 4 membranes-11-00042-t004:** Comparison of experimental results with other membranes.

Membrane Materials	Oil Rejection, %	Water Flux, L/m^2^ h	Reference
Polystyrene	96	230	[39]
Polysulfone	94.21	100	[40]
Polysulfone/Polyvinyl pyrrolidone	99.7	43.6	[41]
Polysulfone/Cellulose acetate/Polyethylene glycol	80	700	[25]
Polyamide	60	100	[42]
Polysulfone/Cellulose acetate	60	40	[43]
Polysulfone	140	-	[44]
Polysulfone/Iron acetate	170	-	
Polysulfone/Polyamide film	310	-	
Polysulfone/Iron acetate/Polyamide film	380	-	
Cellulose acetate/Nylon 66/Dimethyl Sulfoxide (D1)	89	33	This work
Cellulose acetate/Nylon 66/Formic Acid (F3)	95	23	This work

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
