# Peer review of "Development of Polymeric Membranes for Oil/Water Separation"

_membranes, 2021, doi:10.3390/membranes11010042_

Round 1
Reviewer 1 Report
Back to my previous comments, I am not sure if the paper can be accepted by the current version.
- For the first question about Fig. 5 and Table 3. I remember in the first version, their data had conflict between Fig. and Table. I dont know what does authors response mean.
- For the third question, I am not sure if authors only present the 3% Nylon sample can draw the conclusion that upto 20% loading can improve the thermal stability.
- I am not sure how I can make it more clear. Can authors present all the data in series. Can authors have Figure 2 including F1, F2, F3, C1. Can you have Figure 3 including D1, D2, D3, C1. Can authors have Figure 4 including C1, D1,D2,D3, F1,F2,F3.
Author Response
Dear Respected Reviewer,
Greetings
Please find attached the authors' response to your valuable comments.
Thanks with best regards

Reviewer 2 Report
The paper still need some corrections, the authors should consider.
- Title should replace with “developed” instead of synthesized and among the manuscript and the title as well.
- In line 84: the polymer name is “Polyether sulfone (PSF)” change to” Polysulfone “ similarly in table.3
- In mechanical test still need to mention membrane thickness, and how the authors calculate the different mechanical properties which are different in there test? tensile strength and elongation break?
- In figure.5 author change quality, however, STDV error bar were missed in the updated figure.
- Its better to add SEM images of the different membranes with similar scale bar.
Author Response

(The authors gave the same response as above.)

Reviewer 3 Report
The changes have been made satisfactorily and I can recommend publication.
Author Response
Thanks with best regards

This manuscript is a resubmission of an earlier submission. The following is a list of the peer review reports and author responses from that submission.
Round 1
Reviewer 1 Report
I would prefer to reject the paper for a couple reasons.
First of all, the authors do not address the questions from reviewers properly, for example. They removed Table 3 instead of answering the questions that we have to address.
Secondly, if I go through the Table they provided for the state-of-art membranes for comparison. They are very old and most of them do not have water flux, which means either no one cares about the topic (I don't think so), or they have some issue on selecting the reference.
Thirdly, I saw a couple questions related to the properties of the membrane, the authors provided the explanation without any experimental data support. eg. I asked what the different loading of Nylon 66 affect the thermal stability. They said " Up to 20% loading can improve thermal stability of membrane." So I still dont know how the different loading of the Nylon66 affects the thermal stability. I barely see the author prefer to answer the questions not from their own data.
In addition, authors don’t provide data in a series, eg, TGA, only two curve are presented, I am not sure if that can represent all their 6 samples.
Reviewer 2 Report
- Introduction should incorporate and strength the review with some important work in the membranes for oil/water separation for other materials and functions:
- Nature Communications volume 3, Article number: 1025 (2012)
- Environmental Nanotechnology, Monitoring & Management, Volume 14, December 2020, 100314.
- Polymers 2020, 12(11), 2597
- What is the width of the tensile test sample?
- Commercial membrane (C1) should have SEM cross section to show all the comparison between the developed membranes and commercial one.
- Figure .5 quality is not good, replace with higher quality image.
- Mechanical behavior curve is more representative to show in terms of (stress & strain).
- There is an important factor authors recommended to measure which membrane wettability by means of contact angle please check the referred reference (https://doi.org/10.1039/C3TA13397D).
- Results should write with more deeply discussion: for example, “a major drop in the TGA curve is due to the decomposition of CA which is in agreement with the published work [21].” This need more discussions to clarify to the readers.
- All the results reported be compared in table.3 to establish the superiority of the work.
- Figures labels need to be rewritten in better way.
- Whole manuscript. English should be polished in the revision. Several parts have the logic problem. Please be carefully consider English correction.
Reviewer 3 Report
The authors present successful oil/water filtration membranes, with additional analysis including SEM, flux, and robustness testing. The separation effectiveness is shown to be in-line with other reports in the literature. Although the work reports a membrane with a good level of separation performance, it is unclear where the novelty of the work originates, this would need to be emphasised in any revisions of the manuscript. As a result I would not recommend acceptance of the manuscript until this is introduced, in addition to the following aspects:
- Details of the mechanical testing experiments should be provided - to demonstrate how these were performed.
- Details on how the membranes were cross-sectioned should be added, and comment on if this affects the perceived internal structure.
- The authors state that the fabricated membranes were porous, analysis of porosity should be included, or alternative assessment of porosity provided. Not currently clear from evidence provided.
- In table 3 the authors present a comparison with the literature, I would encourage substantive comment on the validity of comparing these literature examples. For example, are they all tested using the same methods, or are there difference that should be noted.
Minor Corrections:
- Section 3.1 is incorrectly labelled 3.7
- The acceleration voltage used in the SEM imaging should be included in the experimental.
- Figure 4 image quality is poor and needs to be improved (resolution increase, thickness of lines, and text size).